# Substrate Specificity of ABCB Transporters Predicted by Docking Simulations Can Be Confirmed by Experimental Tests

**DOI:** 10.3390/molecules29225272

**Published:** 2024-11-07

**Authors:** Mario Röpcke, Sha Lu, Cäcilia Plate, Fee Meinzer, Antonia Lisiecki, Susanne Dobler

**Affiliations:** Institute of Cell and System Biology of Animals, Universität Hamburg, 20146 Hamburg, Germany; sha.lu@uni-hamburg.de (S.L.); caecilia.plate@uni-hamburg.de (C.P.); fee.meinzer@uni-hamburg.de (F.M.); antonialisiecki@posteo.de (A.L.)

**Keywords:** ATP-binding cassette (ABC) transporter, large milkweed bug, *Oncopeltus fassciatus*, cardenolides, co-evolutionary relationships, ATPase activity assays, substrate specificity, Vina, virtual ligand screening, docking simulations

## Abstract

ATP-binding cassette (ABC) transporters, particularly those of subfamily B, are involved in cell detoxification, multidrug resistance, drug treatment pharmacodynamics, and also ecological adaptation. In this regard, ABCB transporters may play a decisive role in the co-evolution between plants and herbivores. Cardenolides, toxic steroid glycosides, are secondary plant metabolites that defend plants against herbivores by targeting their sodium–potassium ATPase. Despite their toxicity, several herbivorous insects such as the large milkweed bug (*Oncopeltus fasciatus*) have evolved adaptations to tolerate cardenolides and sequester them for their own defense. We investigate the role of two ABCB transporters of *O. fasciatus* for the paracellular transport of cardenolides by docking simulations and ATPase assays. Cardenolide binding of *Of*ABCB1 and *Of*ABCB2 is predicted by docking simulations and calculated binding energies are compared with substrate specificities determined in ATPase assays. Both tested ABCB transporters showed activity upon exposure to cardenolides and Km values that agreed well with the predictions of our docking simulations. We conclude that docking simulations can help identify transporter binding regions and predict substrate specificity, as well as provide deeper insights into the structural basis of ABC transporter function.

## 1. Introduction

ATP-binding cassette (ABC) transporters form a highly diversified protein superfamily that plays a fundamental role in the regulation of paracellular transport for a wide range of substrates and occurs in all kingdoms of life. Transporters of the subfamily B (ABCB, P-glycoproteins or MDR proteins) play a special role with regard to the extrusion of xenobiotics [1,2,3]. They can be characterized based on structural features in protein topology: in the functional protein, two membrane-spanning transmembrane domains (TMDs) alternate with two intracellular nucleotide binding domains (NBDs), yielding a TMD1-NBD1-TMD2-NBD2 structure [4]. According to the ATP switch model, paracellular membrane transport is based on a steric change in the TMDs, driven by the hydrolysis of two ATP molecules after dimerization of the NBDs, and results in the extrusion of specific substrates [5]. Since their wide substrate spectrum includes xenobiotics, these potent transporters have a major influence on cell physiology, especially with regard to detoxification mechanisms and the development of multi-resistance [6,7,8,9]. In addition to pharmacodynamic effects in drug treatments, ABCB transporters play an essential role in the adaptation of animals to specific ecological niches or co-evolutionary interactions with their food source [10,11,12,13,14].

Cardenolides are a class of steroid glycosides, which are produced by numerous plants for defense against herbivores [15]. These toxic secondary metabolites can be structurally divided into an aglycone, consisting of a tetracyclic steroid skeleton with a lactone ring, and one or more glycosidically bound sugar residues [15,16]. The toxic effect of cardenolides is based on the inhibition of the sodium–potassium ATPase, which disrupts the cellular ion balance and can lead to cell damage and cell death [17,18]. In addition, the majority of cardenolides are non-polar, which enables them to permeate membranes passively [19]. Cardenolides therefore form an extremely potent defense with which plants can defend themselves from predators.

Despite the highly toxic potential of these secondary plant metabolites, long-term associations between herbivores and their cardenolide-containing food plants have led to coevolutionary relationships [20,21]. The large milkweed bug (*O. fasciatus*, Lygaeidae), a widespread species in North America, has specialized on milkweed plants (*Asclepias* species), which have a high content of cardenolides [22]. The bugs spend their entire life cycle on these plants and even prefer to feed on the particularly toxic seed capsules. In addition to their tolerance for the toxins, the bugs are able to accumulate cardenolides in dorsolateral storage compartments and load them into their defensive secretion [23,24,25]. While their tolerance mechanism for cardenolides has been well studied (target-site insensitivity of the Na,K-ATPase), the trafficking of cardenolides in the bug is the focus of our current research. Since a targeted and active transport away from sensitive organs and into the storage compartments can be assumed, we focus here on ABCB transporters, for which evidence for their involvement in sequestration is accumulating [12,26,27,28].

Testing the substrate spectra of individual ABC transporters can be carried out experimentally with heterologously expressed proteins, yet this procedure is time- and labor-intensive and in the case of cardenolides, the compounds present in the host plant are mostly not commercially available. Docking simulations, on the other hand, are in silico methods for predicting the interaction between a molecule and a protein, which make it possible to determine how substrates might interact with the active binding site of the transporter [29]. The calculated binding energy is a measure of the strength of the interactions between the ligand and the transporter. A higher binding energy indicates a stronger and more specific binding of the substrate to the transporter and allows conclusions about the structural basis of substrate specificity. Furthermore, docking simulations can be used to provide insights into the structural features responsible for binding and recognition of specific substrates by the transporter. Overall, the integration of docking simulations into experimental analyses enables a comprehensive investigation of the relationship between the observed substrate specificity of transporters and the binding energies derived from docking simulations. 

In this study, we use docking models of cardenolides to two ABCB full transporters of *O. fasciatus*, OfABCB1 and OfABCB2, to identify the cardenolide binding regions in these transporters and to compare the calculated binding energies across substrates and transporters in order to predict their substrate specificity. By comparing these simulations with experimentally obtained substrate specificity data from ATPase activity assays of the heterologously expressed proteins, we are able to validate the docking models and confirm the predicted specificities. Our results support docking simulations as a valuable tool to identify substrate-binding regions of ABC transporters and predict the substrate spectrum of these highly versatile transporters.

## 2. Results

### 2.1. Cardenolide: ABCB Docking Simulations

#### 2.1.1. Preliminary Quality Assessment of ABCB Docking Simulation

Virtual ligand screenings are primarily used in pharmaceutical research, particularly in the search for inhibitors. We here used the Vina algorithm to see whether it is able to predict the binding probability of substrates and co-substrates of ABCB transporters. To test the algorithm’s ability to predict molecular interactions in ABCBs correctly, we first redocked ATP to the electro-microscopically (EM) determined structure of the human ABCB4 transporter (Figure 1A). Of the nine models generated by the docking simulation, six models match the actual N-terminal ATP-binding site and three the C-terminal one (Figure 1B). All models showed RMSD (root mean standard division) values less than four for the chosen ATP-binding site, which supports high congruence (Table 1). This could also be confirmed at the amino acid level, where a large part of the correct molecular interactions was found in the models (Figure 1C,D). This shows that the molecular docking of a highly charged molecule, such as ATP, is possible with high accuracy. The fact that docking simulations are able to rediscover the EM-confirmed amino acid interactions was subsequently used in heterologous modeling.

#### 2.1.2. General Considerations About Homology Modeling Procedures

Two approaches are currently available to generate three-dimensional structure predictions for an unknown protein: homologous modeling based on physically verified protein structures or artificial intelligence-based methods as implemented in the popular structure prediction algorithm AlphaFold [30]. We here followed the approach of searching for highly similar proteins with physically verified structures available in the protein database (PDB) to generate homology models for our sequences of interest via the SWISS-MODEL platform. Although models generated with AlphaFold2 are praised for their ever-increasing accuracy [31], warnings have been issued not to use these structures for protein–ligand docking simulations as these artificial intelligence based predictions performed consistently worse in such analyses compared to experimentally determined structures [32,33,34,35,36]. Currently, structure predictions generated on the AlphaFold3 server bear the explicit disclaimer that they should not be used with other docking or screening tools or related technology for biomolecular structure prediction (https://alphafoldserver.com/about, (accessed on 14 October 2024)). As we intended to use the generated structures in the next step for docking simulations of the cardenolides of interest, and these are not available for docking directly on the AlphaFold3 server, we did not generate structure predictions of the OfABCB transporters with AlphaFold and discarded those templates proposed by SWISS-MODEL that resulted from AlphaFold predictions. 

#### 2.1.3. Homology Modeling of the OfABCB1

The three-dimensional structure of OfABCB1 was predicted with the SWISS-MODEL platform using homologous modeling [37]. Nine templates favored by the SWISS-MODEL ranking were selected based on the Global Model Quality Estimate (GMQE) score, which considers the properties of the template–target alignment, the structure of the template, and the Quality Mean Distance constraints global score (QMEANDisCo) of the generated model (Table 2). *Caenorhabditis elegans* pgp-1 (Protein Data Bank ID: 4f4c) turned out to be the most favorable template for the project. The model evaluation of the SWISS-MODEL platform showed that the global quality of the spatial model generated for OfABCB1 fulfilled the criteria for a docking simulation application, with a MolProbity of 1.65 (as low as possible, suitable for docking < 2.0) and a QMEANDisCo global score of 0.67 +- 0.05 (target: 1–0.8, suitable (0.6–0.8) (Figure 2A). 

The Ramachandran plot, which determines the steric quality of the amino acid backbones in protein models, showed that 94.42% of the amino acids lie in the favored region (darker green areas) and only 1.21% appear as Ramachandran outliers (white area) (Figure 2B). This suggests that our model is a well-folded protein structure. In addition, the generated model was examined by the web service ProSA-web, which analyzes protein structures for their biological plausibility based on steric, non-pair interactions and the orientation of the side chains [38]. In the Overall Model Quality Plot, an aggregated assessment of the structural quality of the protein was carried out (Figure 2C). Here, the OfABCB1 model showed a Z-score of −13.3, which corresponds to the average of proteins of similar size in the reference database. The local analysis showed that the ATPase subunits have a lower knowledge-based energy level than the permease subunits of the OfABCB1 model, which indicates their higher conservation (Figure 2D). In summary, we conclude that the three-dimensional model of OfABCB1 has a high structural and biological integrity.

In the context of a docking simulation of large proteins, such as the ABCB transporters, a global structure assessment alone is not sufficient to determine the optimal configuration. Instead, a local approach to identify potential docking regions is necessary. Due to their ability to adopt a conformation spanning the membranes multiple times, ABCB transporters form a transport cleft that is supported by twelve alpha helices that are grouped into two transmembrane domains. In the docking simulations, it can be assumed that cardenolides bind within the generated cavity. A detailed analysis of the two TMDs with regard to quality, confidence, and template similarity is presented in Figure 3. All alpha helices demonstrate a high level of confidence and similarity to the template, indicating that these domains are of high quality and that the model can be utilized for docking experiments.

#### 2.1.4. Homology Modeling of the OfABCB2

The three-dimensional structure of OfABCB2 was likewise created using homologous modeling using the SWISS-MODEL platform [37]. The nine templates favored by the SWISS-MODEL ranking based on the Global Model Quality Estimate score (GMQE), the structure of the template, and the QMEANDisCo global score of the generated model are shown in Table 3. The human bile salt export pump ABCB4 (6lr0) turned out to be the most favorable template for the project. It is noteworthy that this represents the ninth-best result when only evaluating the GMQE score. A0A7E4RIL2 was not selected for being generated by AlphaFold2 and was described as “most likely obsolete”. The templates 6qex, 6fn4, 8pmj, 4q9k, and 4xwk were not selected because they are substrate-complexed protein structures and thus not in the apo state. This may result in the binding regions being inaccessible for docking simulations. The template 4m1m was excluded due to a poor MolProbity Score, which exceeds the project’s cutoff set to 2.0. This leaves 4f4c and 6lr0. To assure the independence of our docking approaches, a different template was chosen for OfABCB2 than for OfABCB1, and therefore 6lr0 was used in the following. The model evaluation of the SWISS-MODEL platform showed that the global quality of the spatial model generated for OfABCB2 was sufficient for a docking simulation application, based on a MolProbity of 2.02 (as low as possible, suitable for docking < 2.0) and a QMEANDisCo global score of 0.66 +- 0.05 (target: 1–0.8, suitable (0.6–0.8) (Figure 4A). The Ramachandran plot showed a ratio of 89.71% amino acids in favored regions with 1.98% outliers. Values of >90% are referred to as excellent, so that very good structural integrity can be assumed here (Figure 4B). In the overall model quality plot, the OfABCB2 model showed a Z-score of −12.17, which is close to comparably sized proteins from the reference database (Figure 4C). Similar to OfABCB1, the higher integrity of the highly conserved ATPase subunits can be seen in the local model quality plot (Figure 4D). Like OfABCB1, OfABCB2 shows high global biological and structural integrity. 

A comprehensive examination of the two transmembrane domains of the OfABCB2 model forming the transport cleft relevant for cardenolide binding is illustrated in Figure 5. The alpha helices as a whole demonstrate a high level of confidence and template similarity, indicating that these domains are of high quality and that the model can be used for docking experiments.

#### 2.1.5. Cardenolide: ABCB Transporter Docking Simulations

The docking simulations were carried out with five cardenolides, which resulted in nine models per ligand (Table A1 and Table A2). Here, only the common aglycon structure was monitored to determine their congruence, ignoring the sugar residues and individual substituents on the steroid core (Figure 6A,B). This procedure led to 45 docking models per ABCB transporter, which were all compared with each other by a heat map (Figure 6C,D). This allowed for identification of the constellations of individual docking models for the five cardenolides that resulted in the highest steric congruence (Table A3). Assuming that each transporter has a unique receptor site for cardenolides with similar amino acid interactions, the model constellations were next examined at the amino acid level, taking all structural features of each cardenolide into account. This demonstrated that constellation no. 40 for OfABCB1 and no. 15 for OfABCB2 showed the greatest similarities in amino acid interactions with the highest degree of congruence between the cardenolide docking models for each individual transporter (Table 4).

#### 2.1.6. Prediction of Putative Active Binding Sites and Structure Analysis

To determine a putative cardenolide binding region in ABCB transporters of *O. fasciatus*, the AA interactions of all 90 generated models were examined considering all structural features of the ligands. In addition to being rather non-polar, the aglycone of cardenolides, consisting of a tetracyclic steroidal core and a lactone ring, forms few H-bonds. For both proteins, the docking simulations showed that the oxygen atoms, especially the carbonyl oxygen atom of the lactone ring, preferentially form bonds with lysine and glutamine residues (ABCB1: Q880, Q955, K214; ABCB2: K237, K737, K743). Furthermore, in the preferred dockings, increased non-polar interactions of mainly alanine and phenylalanine residues (ABCB1:A888, F361, F943; ABCB2: A735, F391, F384) with the steroid framework were observed. Additional bonds were formed with the glycone and cardenolide-classifying modifications of the aglycone.

In the graphical evaluation, the set of selected docking models shows that they cluster at a highly similar position on the two proteins, which is located deep in the gap formed by the membrane helices of the permease subunits (Figure 7). Overall the docking simulations show a very congruent alignment of the five cardenolides within the two ABCB transporters.

A more detailed analysis of the putative docking regions at the level of amino acid interactions reveals that (in the selected docking models) the cardenolides exhibit a significant interaction with both transporters (Figure 8A). With regard to the ratio of hydrogen bonds to alkyl bonds, ABCB1 (H/A bonds: 0.53) exhibits a higher value than ABCB2 (H/A bonds: 0.26). A detailed examination of the alignment of both transporters reveals that the binding regions for ouabain and digoxin—for which we also present experimental data below—are not identical in OfABCB1 compared to OfABCB2. Nevertheless, the binding regions of these two cardenolides within a transporter exhibit a high degree of similarity (Figure 8B). For example, in the five docking models of OfABCB1, a predominance of the amino acids V213, K214, and A888 is evident. A888 forms pi–alkyl bonds to the C or D rings, while V213 and K214, in combination, also form pi–alkyl bonds or form a bond to the carbonyl oxygen atom of the lacone ring (see Figure A1). Similarly, the docking models for OfABCB2 demonstrate that Y244 frequently forms bonds to hydroxyl groups in close proximity to the C ring of the steroid scaffold.

### 2.2. Functional Assays with Heterologously Expressed OfABCB1 and OfABCB2

The coding sequences of OfABCB1 and OfABCB2 were expressed in Sf9 cells by baculovirus infection and the produced proteins harvested as cell membrane preparations. Western blot analysis and immunocytochemistry confirmed the proteins’ correct size and membrane location. The vanadate sensitive activity assays revealed a strong activation of OfABCB1 by digoxin with an estimated maximal velocity (Vmax) of 21.08 ± 1.40 nmol pi/mg protein/min, while ouabain stimulated significantly less activity with a Vmax of 11.78 ± 5.12 × 10^−15^ nmol pi/mg protein/min (Figure 9A; ANOVA F_1,8_ = 14.68, *p* = 0.005). OfABCB2 was significantly less stimulated by digoxin, with an estimated Vmax of 12.54 ± 0.25 nmol pi/mg protein/min and a marginally higher activity stimulated by ouabain with a Vmax of 13.30 ± 0.63 nmol pi/mg protein/min (Figure 9B; ANOVA F_1,8_ = 3.11, *p* = 0.116). A comparison of the full Michaelis–Menten curves confirmed that OfABCB1 is more strongly stimulated by digoxin than by ouabain (F_2,14_ = 23.59, *p* < 0.0001). In a comparison of the transporters, OfABCB1 is more strongly activated by digoxin than OfABCB2 (F_2,14_ = 35.05, *p* < 0.0001), whereas ouabain stimulated OfABCB2 more strongly than OfABCB1 (F_2,14_ = 9.21, *p* = 0.003). 

The K_m_ values of OfABCB1 were higher for both substrates (digoxin 33.73 µM; ouabain K_m_ 48.56 µM), while OfABCB2 already reached half-maximal velocity at much lower concentrations (digoxin K_m_ 9.33 µM; ouabain 7.05 µM). 

### 2.3. Comparison of Calculated Docking Scores and Substrate Specificity

The comparison of these experimentally determined K_m_ values shows good congruence with the prediction from the docking simulations. OfABCB2 compared to OfABCB1 has a delta K_m_ value of −35.51 µM and a delta docking score of −0.8 kcal/mol for ouabain and −24.40 µM and −0.7 kcal/mol for digoxin (see Table 5). Both data sets show that OfABCB2 tends to have a higher affinity for ouabain and digoxin than OfABCB1 although the maximal velocity of OfABCB1 for digoxin is higher. 

## 3. Discussion

ABC proteins are extremely versatile transporters that have played a significant role in the adaptation of insects to toxic compounds in their environments [12,26,41]. The adaptation of insects to cardenolides in their host plants is tightly linked to transport processes that either prevent an uptake of the toxins across the gut membrane or enable their compartmentalization and storage in specific tissues [13,42]. Experimental approaches to determine the involvement of candidate ABC transporters and their specificity for the relevant plant compounds are limited by the availability of these compounds, most of which must be isolated from the plants [12,43,44,45]. Docking simulations provide an alternative for overcoming this limitation.

In the majority of publications, docking simulations on ABC transporters have been limited to identifying inhibitors for improving drug treatments [46,47]. Nevertheless, in silico docking experiments were used to obtain new information about their function and its substrate spectrum, e.g., on human ABCB1 as a vitamin D carrier or the determination of Leishmania ABCB3 as a factor in heme and cytosolic iron/sulfur clusters biogenesis [48,49]. Our use of docking simulations to predict the interaction between relevant compounds and ABCB transporters shows great potential to derive educated guesses about the likelihood and strength of interaction between compound and protein. As a first proof of concept, the known interaction between two ATP molecules and the NBDs of the transporters were recovered precisely as they are known from EM structure determination [PDB-ID: 4f4c], corroborating the suitability of our following docking approach.

When pursuing a docking procedure involving large proteins such as ABCB transporters, particular attention has to be paid to the choice of a homologous template. This is limited by the availability of crystal structures of ABCB transporters of sufficient quality for docking simulations due to the inherent challenges associated with the crystallization of transmembrane proteins in their apo-state conformation. The protein pgp-1 of *Caenorhabditis elegans* (4f4c) proved to be particularly well suited as a template for homologous modeling of the *O. fascciatus* ABCB1 transporter. The transmembrane regions could be modeled both globally and locally with a high level of confidence as evidenced by a QMEANDisCo global score of 0.67 ± 0.05. It is noteworthy that a high Δ QMEANDisCo global score of −0.27 ± 0.05 was achieved, which is attributable to the exceptional quality of the crystal structure of the *C. elegans* pgp-1 (0.90 ± 0.05). With regard to OfABCB2, 4f4c also was highly suitable as template. Nevertheless, in order to achieve greater template independence, we used the also well-fitting human bile salt exporter ABCB11 (6lr0) as an alternative for the docking simulations. It is important to note that in the models for both OfABCB1 and OfABCB2, the good match to the template with high confidence levels holds up for the relevant TMD helices forming the substrate cavity (Aller et al. 2009) but does not apply to the N-terminal primary transmembrane helix where the sequence is poorly conserved. Given the absence of evidence for transport of ouabain or digoxin for either *C. elegans* Pgp-1 or human ABCB11, the difference between template and model could also represent an adaptation of *O. fasciatus* to cardenolides. Overall, based on the overall good match in the TMD regions both protein models can be used for docking simulations.

A comparative analysis of the interactions of OfABCB1 and OfABCB2 with the five tested cardenolides revealed the presence of a unique active center in both cases, although its location on the transporters exhibited slight differences. As anticipated, the binding regions are situated within the membrane gap spanned by the transmembrane helices. The observation that the cardenolides, which, except for ouabain, are all non-polar, form a relatively large number of strong interactions suggests the possibility of different binding mechanisms in the two transporters. This hypothesis is further supported by the disparate ratio of hydrogen bonds to alkyl bonds observed (OfABCB1: 0.53; OfABCB2: 0.26).

A more detailed analysis of an alignment of the two ABCB transporters revealed that the detected amino acid interactions for ouabain and digoxin were not congruent. This allows us to conclude that there are two different binding mechanisms in these two transporters. Given the alpha helix configuration of the putative substrate-binding region of ABCB transporters, the amino acids that interact with the cardenolides are distributed across the different helices of both TMDs. It was not possible to identify a contiguous sequence in this case; however, the docking models for ouabain and digoxin demonstrated similar interactions in the same regions of each transporters. Our models also revealed strong similarities between the two transporters in their interactions between specific residues and the ligands. The generally strongly non-polar cardenolide ligands frequently form alkyl bonds between the steroid ring system and lysine and glutamine residues of the binding pocket. In addition, hydrogen bonds between the carbonyl oxygen atom of the lactone ring and alanine or phenylalanine residues are also remarkably common. In the case of OfABCB1, A888 frequently interacts with pi–alkyl bonds to the C or D rings, while V213 and K214 do the same as a duo or target a hydrogen bond to the carbonyl oxygen atom of the lactone ring. Similarly, OfABCB2 demonstrates that Y244 regularly forms bonds to hydroxyl groups in the proximity of the C ring of the steroid skeleton. Overall, a significant number of interactions are formed with the aglycone. 

The bonds of these omnipresent structures apparently play a key role in the binding and transmembrane transport of cardenolides. Similarly, pharmacodynamic studies show that conjugated steroids (e.g., dexamethasone, estrone, DHEAS) belong to the substrate spectrum of the human ABCB transporters [50,51,52]. It can be assumed that in the large milkweed bug adaptation to cardenolide-containing host plants led to a specialization of these transporters to the vast array of cardenolides present in the plant diet. The docking experiments suggest that the aglycone of cardenolides plays a significant role in binding to the active center, whereas the specific aglycone modifications and the sugar residues influence the substrate specificity. This resembles the situation observed for the binding of cardenolides and bufadienolides to the Na,K-ATPase, where the binding characteristics are strongly affected by modifications of the aglycone and sugar residues [43,53,54].

In our modeling approach, we determined the difference in binding energies of the five monitored cardenolides to OfABCB1 relative to OfABCB2 and we used these data to predict the transporters’ substrate specificities. The substrate specificity describes the ability to recognize and bind a specific substrate, whereas the binding energy represents the energy released when a stable complex is formed. These two properties are connected, as high substrate specificities indicate high steric compatibility between the active site and the ligand, which favors the formation of non-covalent interactions. Likewise, the induced fit model describes the steric adaptation of the protein upon substrate binding to maximize the binding energy. Thus, a higher substrate specificity leads to a higher binding energy, which translates to a low Michaelis–Menten constant (K_m_) and a high turnover rate (k_cat_) as indicators of high affinity and efficiency.

To validate the modeling predictions, we experimentally tested the suitability of two commercially available compounds, digoxin and ouabain, as substrates for the two ABCB transporters. In accordance with the higher docking scores, OfABCB2 had higher affinity for both substrates than OfABCB1 and higher affinity for ouabain than for digoxin. Similarly, the lower docking scores of OfABCB1 for ouabain correspond with the lower observed affinity for ouabain compared to digoxin. The determined docking scores are based on the intermolecular interactions and intramolecular ligand tensions, which include torsion energy, steric hindrance, and conformational flexibility. A distinct correlation can be observed between the number of hydrogen bonds formed and the results of the docking simulations. With regard to the formation of hydrogen bonds, it can be observed that OfABCB1 forms a smaller number of hydrogen bonds to ouabain (OfABCB1:OfABCB2: 1:4) and digitoxin (OfABCB1:OfABCB2: 3:5) than OfABCB2. 

These inferences can be extrapolated for cymarin, frugoside and oleandrin. They all possess highly non-polar characteristics, suggesting that they bind to a hydrophobic active center of the proteins, forming hydrophobic interactions. Yet, the results of the docking simulations demonstrate that ABCB transporters are capable of forming a substantial number of hydrogen bonds, exhibiting considerably higher bond strength. This evidence suggests that polar interactions may be more relevant in the analysis of putative substrate specificities than weak interactions such as alkyl bonds. Since we observed agreement between modeling predictions and experimental results for two of the five substrates, we can extrapolate the behavior of the other three. Thus, we expect OfABCB1 to show a higher substrate specificity for oleandrine and cymarine and no difference between the two transporters in substrate specificity for frugoside. 

We are only starting to investigate the specificity of insect transporters for certain cardenolides experimentally and so far, such experiments are mostly limited to commercially available cardenolides, like ouabain and digoxin [12,24]. The docking simulations presented here provide a highly valuable tool to derive predictions for other compounds, though we also see a need for further experimental validation of these predictions. 

## 4. Materials and Methods

### 4.1. Retrieval of O. fasciatus ABCB Full Transporter Sequences

RNA sequence data obtained from whole bodies of *O. fasciatus* [55] were assembled by Trinity [56] as described in [57]. A further transcript assembly was conducted by CLC Genomics Workbench 7.0.3 (Qiagen, Hilden, Germany). Homology searches by tblastn with previously identified ABCB transporters (MDR proteins) of human origin (2) and various insects (*Drosophila melanogaster* (3), *Bombyx mori* (3), *Tribolium castaneum* (2), *Apis melifera* (1), *Acyrthosiphon pisum* (2), *Lygus hesperus* (1)) yielded two candidates’ transcripts for full transporters, *Ofabcb1* and *Ofabcb2*. Comparison to the *O. fasciatus* genome enabled further 5′ extension of *Ofabcb1.* Analyses by Pfam confirmed the conserved structure of alternating TMDs and NDBs and RT-PCR followed by sequencing could confirm the full cds of *Ofabcb1* of 3849 bp. In contrast, the full cds of *Ofabcb2* could not be retrieved from the transcriptome assemblies but remained partial at the 5′ end where at least one expected transmembrane passage was missing.

To obtain the full coding sequence of *Ofabcb2*, an additional long-read transcriptome was generated. RNA was extracted from thorax wall preparations of *O. fasciatus*, as previous analyses indicated that the gene is strongly expressed in this tissue. RNA was extracted by using the Crystal RNAmagic kit (Machery- Nagel, Düren, Germany) and removing genomic DNA with the gDNA eliminator columns of the RNeasy Plus kit (Qiagen, Hilden, Germany). mRNA libary preparation and sequencing on a PacBio RSII (PacBio, Menio Park, CA, USA) was carried out by Novogene (Cambridge, UK). Raw reads were uploaded to the departmental Galaxy server and searched by blastn using partitions of the previously recovered partial *Ofabcb2* transcript. Retrieved sequences were mapped with Bowtie2 [Galaxy Version 2.3.4.2] [58] to the previous transcript version and visualized and edited with Sequencher (version 5.1; Gene Codes Corp., Ann Arbor, MI, USA). The final assembled sequence had all expected structural motives and the full cds of 3984 bp could be amplified by RT-PCR and verified by sequencing.

### 4.2. Virtual Cardenolide: ABCB Docking Screening

#### 4.2.1. Ligand Preparation

The two-dimensional (2D) structures of five cardenolides, ouabain (PubChem CID: 439501), digoxin (PubChem CID: 2724385), cymarin (PubChem CID: 441853), oleandrin (PubChem CID: 11541511), frugoside (PubChem CID: 120728), and the nucleotide ATP were obtained from the NCBI PubChem database in sdf format. Subsequently, these were converted into mol2 format using the Open Babel software.

#### 4.2.2. Homology Modeling

Homology modeling of the *O. fasciatus* ABC transporter OfABCB1 (1282 aa) and OfABCB2 (1327 aa) were performed using the SWISS-MODEL online software [37]. For OfABCB1, *C. elegans* Pgp-1 (PDB-ID: 4f4c) was employed as a template, while HsABCB11 (PDB-ID: 6lr0) served as the template for OfABCB2. The suitability of the used protein structures and the structural integrity of the models was assessed using Ramachandran plots from SWISS-MODEL and the Protein Structure Analysis web service (ProSA-web) [38].

#### 4.2.3. Molecular Docking 

The molecular docking was performed with the Vina algorithm [59,60], integrated in PrRx 0.8 software [61]. The receptors and ligands were prepared for docking with Discovery Studio (Dassault Systemes BIOVIA) by adding polar hydrogens, assigning Gasteiger charges, and merging non-polar hydrogens using the default algorithm of the program. The suitability of molecular docking simulation was assessed by first redocking ATP molecules to HsABCB4:ATP (PDB-ID: 6s7p). For this, the heteroatoms data sets (HETATM) for ATP, cholesterol, and Mg^2+^ were removed from the pdb file using Discovery Studio. A semi-blind docking was performed, wherein the grid box includes both ATPase subunits. The accuracy of the docking simulation was determined by comparison with the HsABCB:ATP complex. After this proof of concept, virtual ligand screening followed for five cardenolides of interest: ouabain, digoxin, cymarin, oleandrin, and frugoside. According to the translocation activity of ABCB transporters, the grid box was generated to include the permease subunits. Each docking simulation resulted in nine docking models. The exhaustiveness was set to 300 for all molecular dockings. 

#### 4.2.4. Comparison of Generated Models

The comparison of the generated models was carried out using a two-step process. In the first step, the congruence of the models was assessed by determining the root mean standard division (RMSD, Formula (1)). In the case of ATP 31 atoms (excluding hydrogen atoms) and in the case of cardenolides, 25 atoms of the steroid structure and the lactone ring were used.
(1)$$RMSD=\sqrt{\frac{\sum_{i=1}^{N}\left({\left(B_{i,x}-A_{i,x}\right)}^{2}+{\left(A_{i,y}-B_{i,y}\right)}^{2}+{\left(A_{i,z}-B_{i,z}\right)}^{2}\right)}{N}}$$
where the following hold:
*N* = Number of atoms to compare*A*: Docking model A*B*: Docking model B x: x-coordinate *y*: y-coordinate*z*: z-coordinate


In the second step, a heat map was created to check for the congruence of the 45 docking models of the cardenolides and the average RMSD of the best models for each cardenolide was determined. In the second instance, the occurrence and type of amino acid interactions between the ligand and the receptor were compared with each other using the Discovery Studio software. In the case of elucidating a common binding region, the same amino acid interaction in each model composition and the similarity factor of a model set was determined by Formula (2).
(2)$$Sf=\frac{B_{i,similar}}{B_{i,total}}$$
with the following:Sf: = Similarity factor of ligand model constellations based on similar amino acid interactions.*B_i_*_,similar_: = Number of all AA interactions proposed in multiple ligand models.*B_i_*_,total_: = Total number of all AA interactions in the ligand models.

### 4.3. Enzyme Assays with Heterologously Expressed OfABCB1 and OfABCB2

The coding sequences of *Ofabcb1* and *Ofabcb2* were amplified from start to stop codon by RT-PCR of RNA extractions, cloned into pFBD plamids (Invitrogen, Carlsbad, CA, USA), and controlled by Sanger sequencing. Bacmids were generated according to the manufacturer’s protocol (Invitrogen) and transfected with Cellfectin II (Thermo Fisher Scientific, Waltham, MA, USA) into *Sf9* cells. After three days, recombinant baculoviruses were harvested by centrifugation, used to infect a new batch of Sf9 cells with baculoviruses, and harvested once more after five days. A final batch of *Sf9* cells was infected with this P2 virus stock and cells, harvested by ultracentrifugation five days later (for details see [62]). Correct membrane location was verified by immunocytochemistry as in [12]. Cell membranes were isolated from the cell pellets, protein concentrations determined by Bradford assays, and checked for correct protein expression by Western blot as described in [12] (modifications: BSA instead of milk powder and visualization by 50 mM Tris-HCl pH 7.5, 4-chloro-1-naphthol stock solution, 30% hydrogen peroxide in a ratio of 1000:10:1). ATPase assays to determine ABCB activity of the membrane fractions followed the protocols of [12,63] keeping the protein incubation at 37 °C as in the original protocol. Briefly, the phosphate released was quantified at different substrate concentrations in assays that either contained just the tested substrate or additionally the ABCB inhibitor vanadate (500 µM). The difference between the two measured phosphate concentrations corresponds to the activity of ABCB transporters in the membranes. The two commercially available cardenolides ouabain and digoxin (both Merck, Darmstadt, Germany) were tested as substrates at concentrations between 0 and 800 µM in a solution of 2% dimethyl sulfoxide (DMSO) in water. 

All experiments were performed with five biological replicates (independently generated cell membranes) and two technical replicates (identical assays executed in parallel). The differences between non-inhibited and vanadate-inhibited activities were analyzed as dose–response curves and visualized in Origin (version 2021b), which was also used for pairwise statistical comparisons of the data sets.

## Figures and Tables

**Figure 1 molecules-29-05272-f001:**
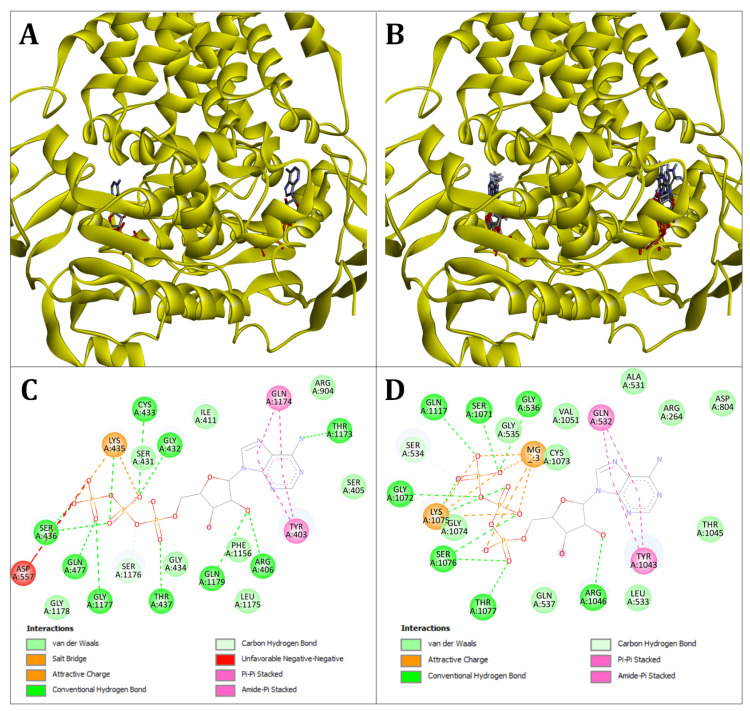
Three-dimensional locations of the ATP in the EM structure of HsABCB4 and newly generated docking models. (**A**) Three-dimensional locations of the ATP to HsABCB4 determined electro-microscopically (EM). (**B**) Three-dimensional locations of the overlaid generated ATP docking models to HsABCB4. (**C**) Amino acid interactions of EM-detected ATP with the N-terminal ATPase subunit. (**D**) Amino acid interactions of EM-detected ATP with the C-terminal ATPase subunit.

**Figure 2 molecules-29-05272-f002:**
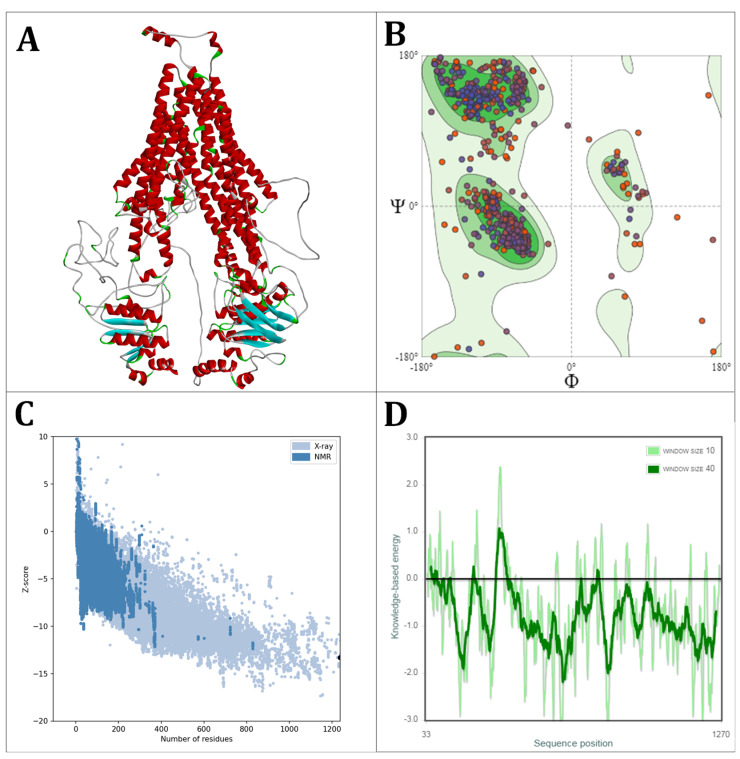
Global assessment of the homology modeling of OfABCB1. (**A**) Three-dimensional model of OfABCB1 obtained with SWISS-MODEL based on the best-fitting available structure. (**B**) Ramachandran plot of OfABCB1 representing the energetically favored regions of the dihedral angles of the backbone relative to the amino acid residues of the protein structure. Here, darker green areas indicate increasing fit with the available structure; the color code of dots gives the similarity of amino acids to the model scaffold, with blue indicating identical, purple similar and red different amino acids. (**C**) Overall Model Quality Plot by ProSA-web of OfABCB1 plotting the Z-scores of the known protein structures against their amino acid chain length (blue) into which OfABCB1 (black dot) is inserted. (**D**) Local Model Quality Plot by ProSA-web of OfABCB1 evaluating the local quality of the protein model, in which the amino acids along the sequence are plotted against the local Z-values, considering the 10 (light green) or 40 (dark green) neighboring amino acids. Low y-values indicate high structural quality, while high Z-values indicate potentially problematic or less reliable areas. The areas with high Z-values contain the transmembrane domains, which represent less conserved regions.

**Figure 3 molecules-29-05272-f003:**
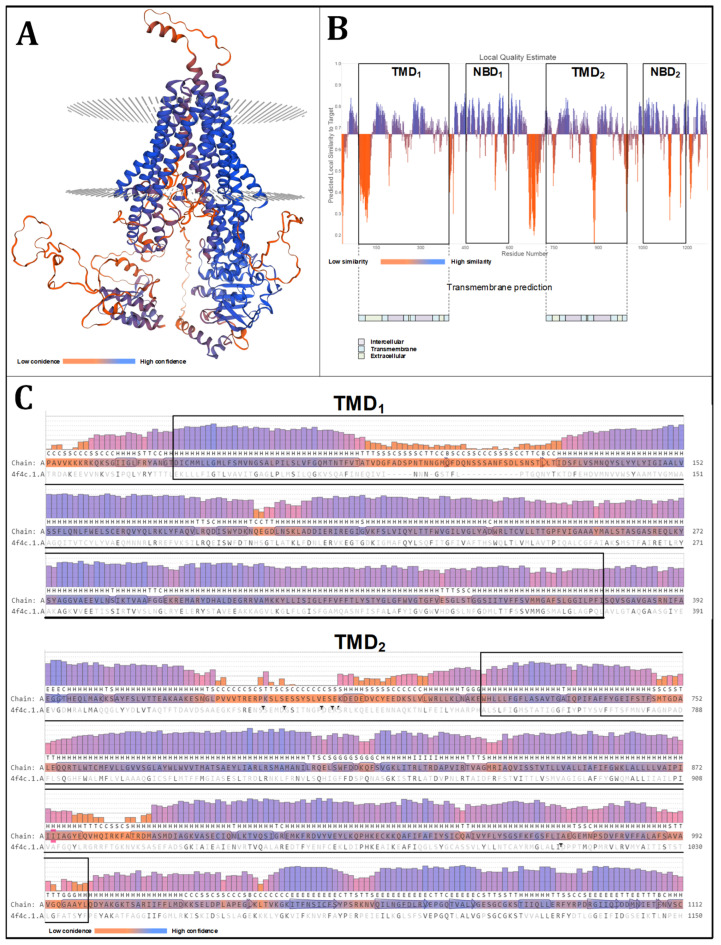
Local assessment of the homology modeling of OfABCB1, focusing on the transmembrane domains (TMDs). (**A**) Three-dimensional model of OfABCB1 obtained with SWISS-MODEL based on the *Caenorhabditis elegans* multidrug resistance protein pgp1 (4f4c). The coloring indicates the confidence level in the obtained model. (**B**) Local quality estimation of the obtained OfABCB1 model on the amino acid level (SWISS-MODEL, model assessment). The distinctive ABCB topology is illustrated (as determined by HMMER) with the transmembrane prediction provided by DeepTMHMM [39,40]. (**C**) Confidence assessment of OfABCB1 on the amino acid level according to SWISS-MODEL assessment, illustrating the relevant regions of the two TMDs.

**Figure 4 molecules-29-05272-f004:**
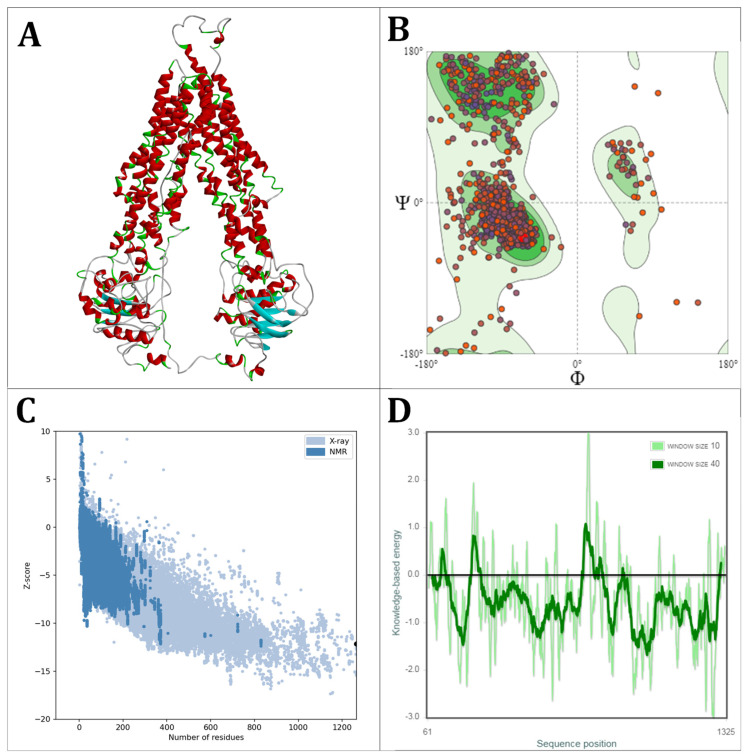
Assessment of the homology modeling of OfABCB2. (**A**) Three-dimensional model of OfABCB1 obtained with SWISS-MODEL based on the best-fitting available structure. (**B**) Ramachandran plot of OfABCB2 representing the energetically favored regions of the dihedral angles of the backbone relative to the amino acid residues of the protein structure. Here, darker green areas indicate an increasing fit with the available structure; the color code of dots gives the similarity of amino acids to the model scaffold, with blue indicating identical, purple similar, and red different amino acids. (**C**) Overall Model Quality Plot by ProSA-web of OfABCB2 plotting the Z-scores of the known protein structures against their amino acid chain length (blue) into which OfABCB2 (black dot) is inserted. (**D**) Local Model Quality Plot by ProSA-web of OfABCB2 evaluating the local quality of the protein model, in which the amino acids along the sequence are plotted against the local Z-values, considering the 10 (light green) or 40 (dark green) neighboring amino acids. Low y-values indicate high structural quality, while high Z-values indicate potentially problematic or less reliable areas. The areas with high Z-values contain the transmembrane domains, which represent less conserved regions.

**Figure 5 molecules-29-05272-f005:**
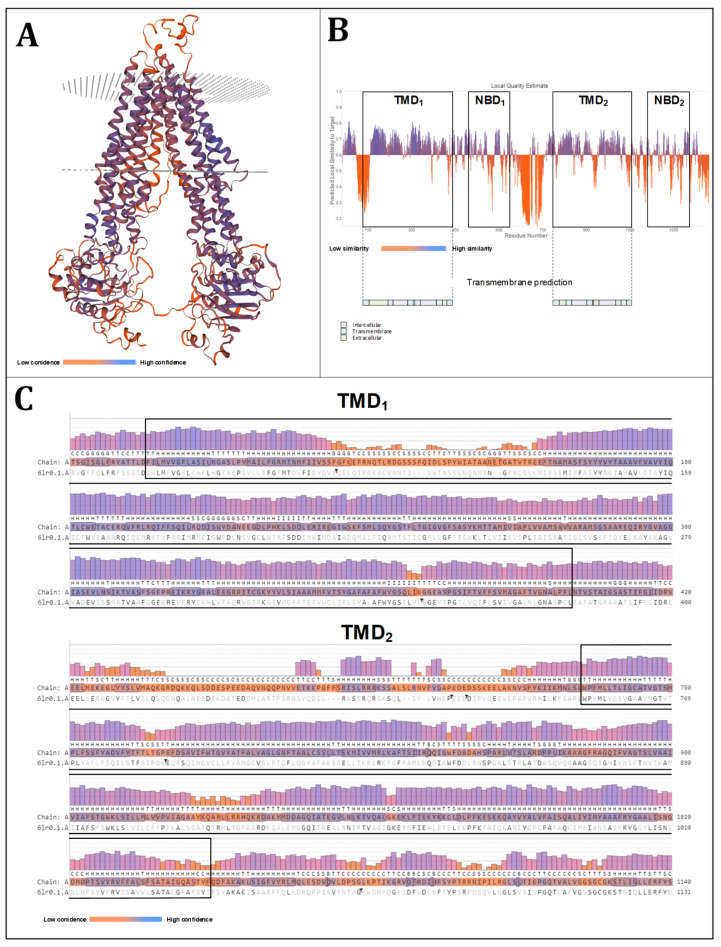
Local assessment of the homology modeling of OfABCB2 focusing on the transmembrane domains (TMDs). (**A**) Three-dimensional model of OfABCB2 obtained with SWISS-MODEL based on the human bile salt export pump ABCB11 (6lr0). The coloring indicates the confidence level in the obtained model. (**B**) Local quality estimation of the obtained OfABCB2 model on the amino acid level (SWISS-MODEL, model assessment). The distinctive ABCB topology is illustrated (as determined by HMMER), with the transmembrane prediction provided by DeepTMHMM [39,40]. (**C**) Confidence assessment of OfABCB2 on the amino acid level according to SWISS-MODEL illustrating the relevant regions of the two TMDs.

**Figure 6 molecules-29-05272-f006:**
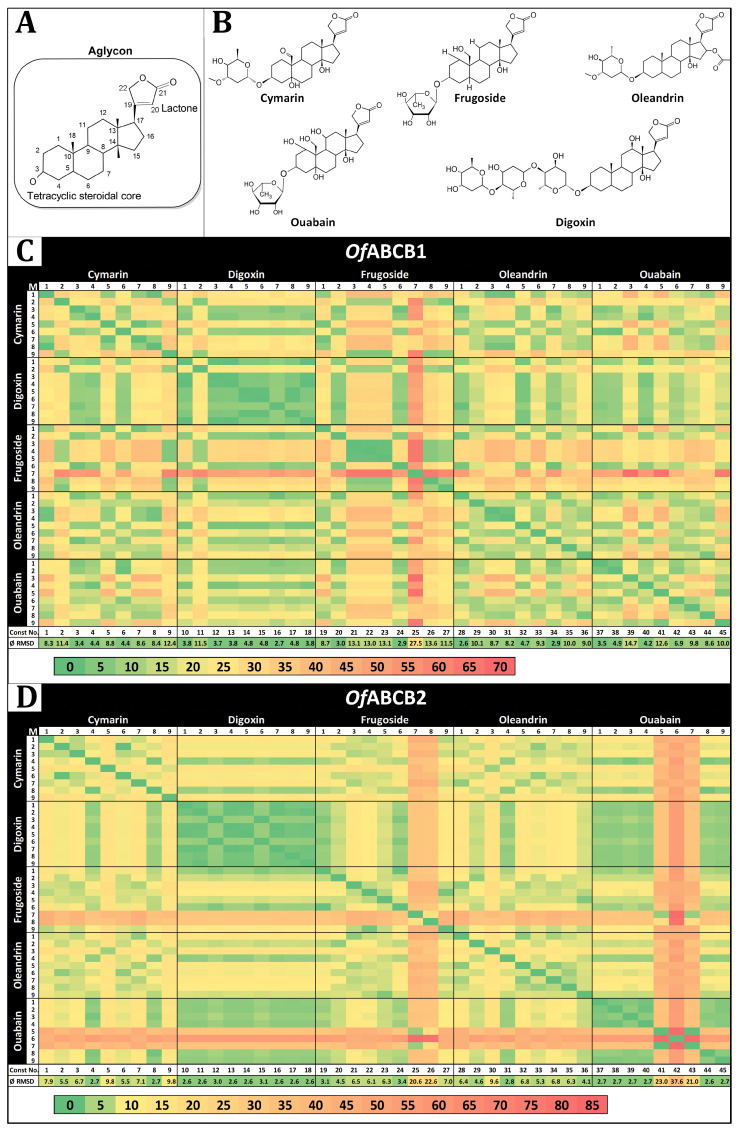
Presentation of the cardenolides and convergence heat maps. (**A**) Common core structure of cardenolides. (**B**) Selected cardenolides used as ligands in the docking simulation. (**C**) Convergence heat map for OfABCB1. (**D**) Convergence heat map for OfABCB2. Const. No.: number of the constellation consisting of one model for each cardenolide of the column, relative to the position of the model of the column; ØRMSD: mean of the RMSD values related to the respective model constellation.

**Figure 7 molecules-29-05272-f007:**
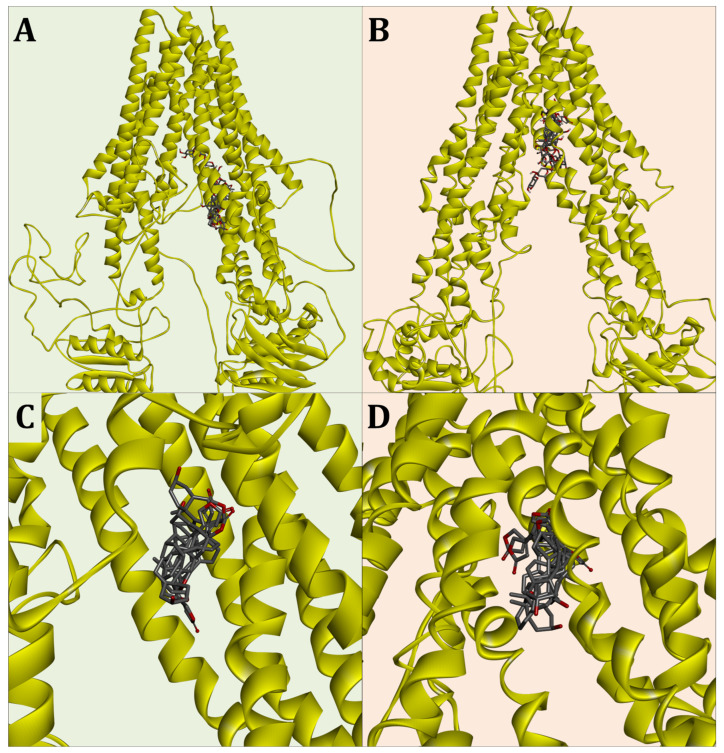
Three-dimensional convergence of the five tested cardenolides in the most suitable docking simulations. (**A**) Merged OfABCB1 cardenolide dockings in model constellation No. 40.; (**B**) merged OfABCB2 cardenolide dockings in model constellation No. 15; (**C**) enlarged docking of just the aglycones for the model constellation No. 40 against OfABCB1; (**D**) enlarged docking of just the aglycones for the model constellation No. 15 against OfABCB2.

**Figure 8 molecules-29-05272-f008:**
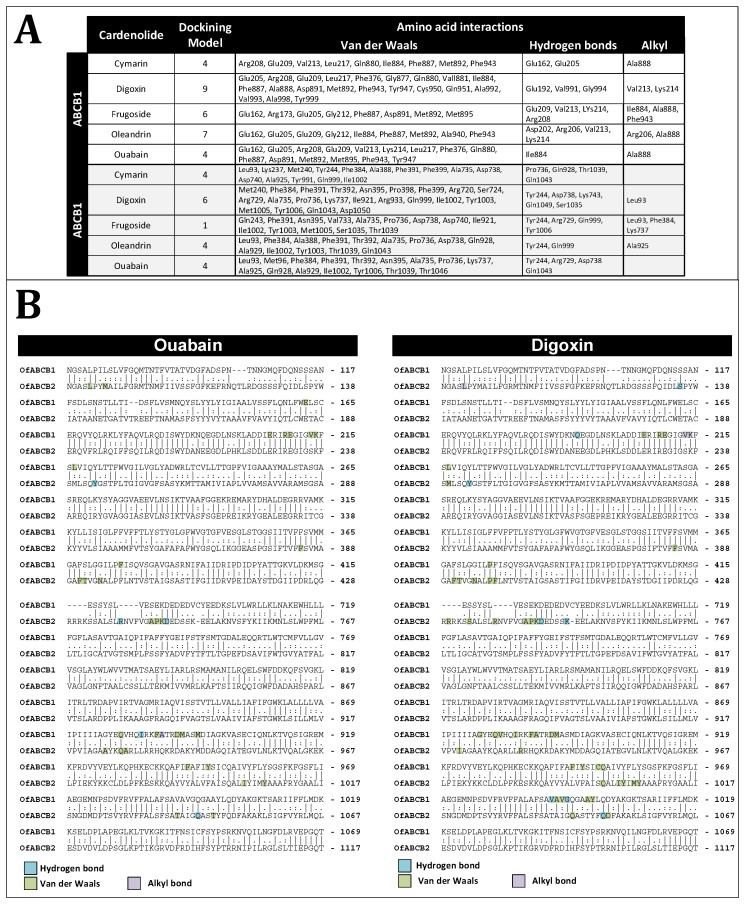
Detailed analysis of the interactions between the five investigated cardenolides and OfABCB1 and OfABCB2. (**A**) Summary of the hydrogen bonds, van der Waals interactions, and alkylic interactions in the docking models for the most favorable constellation. The analysis was conducted using Discovery Studio, Biovia, and a graphical representation was created with Microsoft Visio. (**B**) The illustrated sections are the relevant parts of global alignments of OfABCB1 and OfABCB2 according to the Needleman–Wunsch algorithm, using the Blosum62 matrix, a gap open penalty of 10.0, and a gap extend penalty of 1.0. The total alignment is 1334 amino acids in length and exhibits an identity of 48.73%, a similarity of 67.99%, and gaps of 4.42%. The detected amino acid interactions are highlighted for ouabain and digoxin. The identities and similarities in the alignment are classified as identical (|), similar (:), or dissimilar (.). The graphical representation was created with Microsoft Visio 2019 Version 1808.

**Figure 9 molecules-29-05272-f009:**
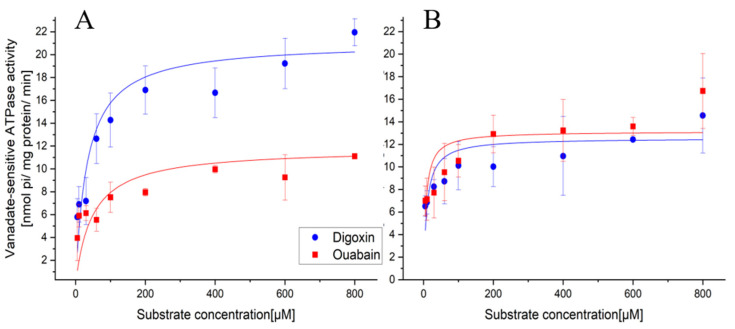
Effects of digoxin and ouabain on OfABCB1 (**A**) and OfABCB2 (**B**) expressed by baculovirus infection in Sf9 cells. The transporter activity is measured as the difference in released phosphate between non-inhibited cell membrane extractions and samples inhibited with 500 µm vanadate.

**Table 1 molecules-29-05272-t001:** Comparison of the root mean standard deviation (RMSD) between EM-detected locations of the generated ATP and ATP-docking models, here showing the spatial congruence between the two. In addition, the predicted amino acid interactions between the ATP molecule at the assumed docking position and the transporter are given. As a single ATP molecule was docked, only one of the two possible ATP-binding regions is modeled.

Ligand	Docking Score	RMSD to ATP1	RMSD to ATP2	H	C–H	SB, AC	π-π, Am-π	UNN
ATP_1	-	0.000	29.291	R406, G432, C433, S436, T437, Q477, T1173, G1177, Q1179	S1176	K435	Y403, Q1174	D557
ATP_2	-	29.291	0.000	G536, R1046, S1071, G1072, S1076, T1077, Q1117	S534, G1072	K1075	Q532, Y1043	-
Model_1	−11.0	2.354	28.801	R406, C433, G434, S436, T437, Q477, G1177, G1178, Q1179	L1175, S1176	K435	-	Q440, D557
Model_2	−10.8	2.775	28.336	R406, S431, G432, C433, G434, S436, Q477, T1173, G1177, G1178	T437, S1176	K435	-	R904, D557
Model_3	−10.8	2.743	28.318	D166, S431, G432, C433, G434, T437, Q477, T1173, G1177, G1178	S1176	K435	-	R904, D557
Model_4	−10.7	2.355	28.434	D166, S431, G432, C433, G434, T437, Q477, T1173, G1177, G1178, Q1179	S1176	K435	-	D557
Model_5	−10.7	29.057	2.230	G535, G536, Q537, R1046, G1072, C1073, G1074, S1076, T1077, Q1177	S534, S1071	K1075	-	Q1080
Model_6	−10.7	2.838	28.252	S431, G432, C433, G434, S436, T437, Q477, T1173, G1177, G1178, Q1179	T437, S1176	K435	-	Q440, D557, R904
Model_7	−10.6	28.378	3.571	G535, G536, G1072, C1073, G1074, S1076, T1077, Q1080, Q1117	S534, S1071	K1075	-	-
Model_8	−10.5	1.430	29.092	C433, S436, T437, Q477, T1173, G1177, G1178	L1175, S1176	K435	-	D557
Model_9	−10.5	28.000	2.152	Q532, G535, G536, Q537, D804, G1072, C1073, G1074, S1076, T1077, Q1117	L533, S534, S1071	K1075	-	-

H: conventional hydrogen bond; C–H: carbon–hydrogen bond; SB: salt bridge; AC: attractive charge; π-π: pi–pi stacked; Am-π: amide–pi stacked; UNN: unfavorable negative–negative.

**Table 2 molecules-29-05272-t002:** The nine highest quality templates for heterologous modeling of *O. fasciatus* ABCB1. Shown are the nine best-fitting templates for heterologous modeling of ABCB1 from *O. fasciatus* on the SWISS-MODEL platform, based on the Global Model Quality Estimate (GMQE) score. The data used to determine the GMQE, such as sequence coverage, sequence identity, global energetic, and structural integrity, as well as the quality of the template and the generated model via the molecular probability and the QMEAN-DisCO Global Scores, are also shown.

Protein ID	Description	Coverage	GMQE	Identity	Method	Template		Model	
						MolProbity Score	QMEANDisCo Global	MolProbity Score	QMEANDisCo Global
4f4c	*C. elegans*, multidrug resistance protein pgp-1	0.96	0.70	41.30	X-ray, 3.4 Å	2.23	0.90 ± 0.05	1.65	0.67 ± 0.05
5kpi	*Mus musculus*, mouse native PGP	0.95	0.69	45.19	X-ray, 4.0 Å	1.43	0.76 ± 0.05	1.42	0.67 ± 0.05
8pmj	*H. sapiens*, bile salt export pump, Vanadate-trapped BSEP in nanodiscs	0.97	0.69	42.34	EM	1.44	0.82 ± 0.05	1.28	0.66 ± 0.05
8pmd	*H. sapiens*, bile salt export pump, Nucleotide-bound BSEP in nanodiscs	0.97	0.68	41.87	EM	1.59	0.81 ± 0.05	1.26	0.66 ± 0.05
A0A2R7W6B8	*O. fasciatus*, uncharacterized protein, most likely obsolete	0.88	0.67	97.44	AlphaFold v2	1.13	0.61 ± 0.05	1.68	0.69 ± 0.05
6fn4	*H. sapiens*, *M. musculus*, UIC2 Fab complex of human–mouse chimeric ABCB1	0.97	0.66	43.59	EM	1.99	0.73 ± 0.05	1.24	0.66 ± 0.05
6s7p	*H. sapiens*, nucleotide-bound phosphatidylcholine translocator ABCB4	0.95	0.66	46.24	EM	1.28	0.75 ± 0.05	1.10	0.67 ± 0.05
6lr0	*H. sapiens*, bile salt export pump, ABCB11	0.97	0.63	42.34	EM	1.90	0.74 ± 0.05	1.81	0.61 ± 0.05
7e1a	*H. sapiens*, bile salt export pump, ABCB11 in complex with taurocholate	0.97	0.60	42.34	EM	2.01	0.74 ± 0.05	1.62	0.60 ± 0.05

**Table 3 molecules-29-05272-t003:** The nine highest quality templates for heterologous modeling of *O. fasciatus* ABCB2, based on the GMQE score. Shown are the nine best-fitting templates for the heterologous modeling of ABCB2 from *O. fasciatus* on the SWISS-MODEL platform, based on the Global Model Quality Estimate (GMQE) score. The data used to determine the GMQE, such as sequence coverage, sequence identity, global energetic and structural integrity, as well as the quality of the template and the generated model via the molecular probability and the QMEAN-DisCO Global Scores, are also shown.

Protein ID	Description	Coverage	GMQE	Identity	Method	Template:		Model:	
						MolProbity Score	QMEANDisCo Global	MolProbity Score	QMEANDisCo Global
A0A7E4RIL2	*O. fasciatus*, uncharacterized protein, most likely obsolete	0.94	0.75	73.21	AlphaFold v2	1.55	0.66 ± 0.05	1.25	0.76 ± 0.05
6qex	*H. sapiens*, ABCB1 in complex with UIC2 fab	0.94	0.63	44.66	EM	1.86	0.74 ± 0.05	1.37	0.63 ± 0.05
6fn4	*H. sapiens*, *M. musculus*, UIC2 Fab complex of human–mouse chimeric ABCB1	0.95	0.63	42.88	EM	1.99	0.73 ± 0.05	1.4	0.64 ± 0.05
8pmj	*H. sapiens*, bile salt export pump, Vanadate-trapped BSEP in nanodiscs	0.95	0.63	41.16	EM	1.44	0.82 ± 0.05	1.16	0.65 ± 0.05
4f4c	*C. elegans*, multidrug resistance protein pgp-1	0.93	0.63	42.74	X-ray, 3.4 Å	2.23	0.9 ± 0.05	1.81	0.65 ± 0.05
4q9k	*M. musculus*, P-glycoprotein 1A co-crystallized with QZ-Leu	0.92	0.62	45.48	X-ray, 3.8 Å	1.29	0.75 ± 0.05	1.31	0.64 ± 0.05
4xwk	*M. musculus*, P-glycoprotein co-crystallized with BDE-100	0.92	0.62	45.48	X-ray, 3.5 Å	1.86	0.76 ± 0.05	1.33	0.64 ± 0.05
4m1m	*M. musculus*, multidrug resistance protein 1A	0.92	0.62	45.57	X-ray, 3.8 Å	2.34	0.75 ± 0.05	1.38	0.63 ± 0.05
6lr0	*H. sapiens*, bile salt export pump ABCB11	0.95	0.59	41.16	EM	1.9	0.74 ± 0.05	2.02	0.66 ± 0.05

**Table 4 molecules-29-05272-t004:** Bond similarity analysis of the cardenolide model constellations. The model constellations were selected in the first step based on congruence (the top 15) and in the second step based on shared amino acid interactions. The similarity factor (sf) determined based on this procedure has a higher priority than the RMSD value of the congruence analysis. The binding scores determined are listed, which represent the calculated binding affinity of all the structural features of each cardenolide ligand of a specific model with the respective transporter. Red boxes mark the chosen model constellations.

Protein	Constellation No.	Mean RMSD	Bonds (Total)	Bonds (Similar)	Sf	Binding Scores [kcal/mol]				
						Cym	Dig	Fru	Ole	Oua
OfABCB1	28	2.6	13	7	0.54	−8.4	−9.8	−8.9	−9.0	−8.5
	16	2.7	13	7	0.54	−8.4	−9.8	−8.9	−9.0	−8.5
	24	2.9	15	8	0.53	−8.4	−9.8	−8.4	−8.8	−8.5
	34	2.9	15	8	0.53	−8.4	−9.8	−8.4	−8.8	−8.5
	20	3.0	13	7	0.54	−8.4	−9.8	−8.9	−8.9	−8.5
	3	3.4	13	8	0.62	−8.4	−10.1	−8.4	−8.8	−8.5
	37	3.5	14	8	0.57	−8.3	−9.8	−8.9	−9.0	−8.5
	12	3.7	16	10	0.63	−8.2	−10.1	−8.9	−8.9	−7.8
	10	3.8	17	10	0.59	−8.2	−10.3	−8.9	−8.9	−7.8
	13	3.8	18	12	0.67	−8.2	−9.7	−8.9	−8.9	−7.8
	18	3.8	18	12	0.67	−8.2	−9.7	−8.9	−8.9	−7.8
	40	4.2	19	13	0.68	−8.2	−9.7	−8.4	−8.8	−7.8
	4	4.4	19	13	0.68	−8.2	−10.0	−8.4	−8.8	−7.8
	6	4.4	14	6	0.43	−8.3	−9.8	−8.9	−9.0	−8.4
	32	4.7	16	10	0.63	−8.2	−10.1	−8.9	−8.9	−7.8
OfABCB2	10	2.6	28	17	0.68	−8.0	−10.7	−8.5	−8.2	−8.3
	11	2.6	29	17	0.63	−8.0	−10.6	−8.5	−8.2	−8.6
	13	2.6	30	19	0.70	−8.0	−10.5	−8.5	−8.2	−8.3
	14	2.6	27	18	0.75	−8.0	−10.4	−8.5	−8.2	−8.3
	16	2.6	29	20	0.74	−8.0	−10.4	−8.5	−8.2	−8.6
	17	2.6	26	17	0.74	−8.0	−10.3	−8.5	−8.2	−8.3
	18	2.6	28	19	0.73	−8.0	−10.3	−8.5	−8.2	−8.6
	44	2.6	25	9	0.41	−7.8	−10.6	−8.5	−8.2	−8.0
	4	2.7	32	19	0.63	−8.0	−10.5	−8.5	−8.2	−8.4
	8	2.7	28	15	0.60	−7.8	−10.5	−8.5	−8.2	−8.0
	37	2.7	25	15	0.65	−8.0	−10.3	−8.5	−8.2	−8.7
	38	2.7	27	13	0.52	−8.0	−10.6	−8.5	−8.2	−8.0
	31	2.8	26	14	0.61	−7.8	−10.7	−8.5	−8.2	−8.0
	12	3.0	30	21	0.75	−8.0	−10.5	−8.5	−8.2	−8.6
	15	3.1	30	22	0.79	−8.0	−10.4	−8.5	−8.2	−8.6

RMSD: root mean standard deviation; Bonds (Total): number of bonds between all chosen docking models and a protein model; Bonds (Similar): number of shared amino acid interactions in the chosen model constellation; Sf: similarity factor based on amino acid interactions; Constellation No. based on the congruence heat map.

**Table 5 molecules-29-05272-t005:** Comparison of docking scores and Km values of OfABCB1 and OfABCB2.

	Docking Score [kcal/mol]		K_m_ Values [µM] ± SE		OfABCB2 Relative to OfABCB1	
Cardenolide	OfABCB1	OfABCB2	OfABCB1	OfABCB2	Δ Docking Score [kcal/mol]	Δ K_m_ Values [µM]
Ouabain	−7.8	−8.6	48.56 ± 6.83 × 10^−17^	7.05 ± 2.61	−0.8	−35.51
Digoxin	−9.7	−10.4	33.73 ± 11.37	9.33 ± 2.43	−0.7	−24.40
Cymarin	−8.3	−8.0	-	-	0.3	-
Frugoside	−8.4	−8.5	-	-	−0.1	-
Oleandrin	−8.8	−8.2	-	-	0.6	-

## Data Availability

Sequences of OfABCB1 and OfABCB2 have been submitted to Genbank Acc. Nos. PQ521210 and PQ521211.

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
