# Peer review of "Substrate Specificity of ABCB Transporters Predicted by Docking Simulations Can Be Confirmed by Experimental Tests"

_molecules, 2024, doi:10.3390/molecules29225272_

Round 1

Reviewer 1 Report

Comments and Suggestions for Authors

The manuscript "Substrate specificity of ABCB transporters predicted by docking simulations can be confirmed by experimental tests" is carefully written and experimentally consistent.

All models generated by docking appear consistent and with RMSD values ​​less than 4, which is highly desirable.

The results show that both OfABCB transporters have a higher affinity for the cardenolides evaluated.

I consider that the discussion and conclusions are not solid enough, they should be improved, for example, why the difference between the affinity for digoxin and ouabain? I think that is something that should be discussed in greater detail based on the docking analysis and the possible interactions between enzyme and ligand.

Author Response

----

Reviewer 2 Report

Comments and Suggestions for Authors

The manuscript presents interesting work, but requires careful revision to be acceptable for publication. The authors tend to overestimate the quality of the results by using emphatic terms and omitting some important aspects that are necessary to justify the quality of the results.

The suitability of the templates and the structural integrity of the models should be evaluated not only by looking at the stereochemical and energetical properties of the modelled structures. The comparison of the template and target sequences must be reported, to evaluate the overall level of similarity as well as local differences at the regions of the binding sites, that may influence the quality of docking simulations. With large proteins, as in this case, the overall quality may be not sufficient to give reliable docking simulations. The comparison of stereochemical properties and energy profiles of both - template and targets - must be presented, sa it may confirm that the quality of the target model is quite similar to that of the template (while if the quality of the target is significantly worse, the template is probably not appropriate). At the same time, at lines 122-123 the affirmation "proved to be the most suitable by Ramachandran plot analysis" referred to the template is not correct, with reference to  "the most". There is no proof that the template used is better than another one, therefore the affirmation "the most suitable" is wrong.

Due to the development of the most recent methods of predictions of the structure of a protein, the Authors should at least compare the models obtained by SwissModel with models obtained by AlphaFold (or others), and justify the use of the SwissModel structure.

The experimental assays do not confirm the most of the structural analysis, because there is no experimental detail about what are the sites of interaction. In more detail, the essays give a confirmation of the ability to bind the two ligands used, but looking at the standard error values, it seems that the confidence interval is not sufficient to confirm the preference for one of the two ligands. The complete computation for high confidence level (95%) should be added, and results appropriately discussed.

The use of "data not shown" should be abolished. It is an old practice from when publication space was limited in terms of figures and tables allowed in the article. Now, no limits to figures and tables, the opportunity to include supplementary materials, as well as to deposit large data sets in public archives, the open science suggestions and the rules for transparency, converge to suggest that all data mentioned in an article as proof of any affirmation must be shown.

Careful revision of the text can also improve minor points and inattention as in:
- line 364: the structures of templates are "crystals" only in one of the two cases.
- line 455: Data Availability Statement is incomplete for the accession numbers.

Round 2

Reviewer 2 Report

Comments and Suggestions for Authors

The authors' response on the comparison I had asked for their model with a model made with AlphaFold is not satisfactory.

AlphaFold is demonstrated to be superior to pre-existing methods, including Swissmodel, in the quality of the models. If the AlphaFold model is not for docking simulations, the same limitations should be considered for the model obtained with SwissModel. Moreover, there are further evolutions of AlphaFold, also suitable for docking simulations.

In view of this, I confirm my request to make a model with AlphaFold and compare it with the model used; it is possible that the two models are very similar, the quality of the models might be different, and this might be in favor of the model used (this will enforce the results obtained by the authors); if not, a discussion must still be presented to justify the use of the model obtained with Swissmodel. I find it essential to discuss and justify in the article why Swissmodel was used, and not AlphaFold, as any reader of the article would expect for a protein modelling step; it is necessary for the reader to be convinced of the quality of the work done by the authors.

Comments on the Quality of English Language

Nothing to report

Author Response

Academic Editor Evaluation Round 1

Thanks for providing an extensively revised version, where experimental data is included with recombinant enzymes and inhibition assays. The major complaint seems to be the disregard for AlphaFold as a method for structure prediction. In my experience, Swiss models could not phase novel crystallographic datasets, whereas AlphaFold predictions could resolve the datasets by molecular replacement.

Please sustain your asseverations with literature or further discussion, which is also a valuable contribution to the literature.

Author's Reply

Dear Academic Editor,
we thought that we have sufficiently addressed this point. Please reread our paragraph l. 121-139:

2.1.2. General considerations about homology modelling procedures
Two approaches are currently available to generate three-dimensional structure predictions for an unknown protein: homologous modelling based on physically verified protein structures or artificial intelligence based methods as implemented in the popular structure prediction algorithm AlphaFold [30]. We here followed the approach of searching for highly similar proteins with physically verified structures available in the protein data base (PDB) to generate homology models for our sequences of interest via the Swiss-Model platform. Although models generated with AlphaFold2 are praised for their ever increasing accuracy [31] warnings have been issued not to use these structures for protein ligand docking simulations as these artificial intelligence based predictions performed consistently worse in such analyses compared to experimentally determined structures [32–36]. Currently structure predictions generated on the AlphaFold3 server bear the explicit disclaimer that they should not be used with other docking or screening tools or related technology for biomolecular structure prediction (https://alphafold server.com/about, Oct. 14, 2024). As we intended to use the generated structures in the next step for docking simulations of the cardenolides of interest, and these are not available for docking directly on the AlphaFold3 server, we did not generate structure predictions of the OfABCB transporters with AlphaFold and discarded those templates proposed by Swiss-Model that resulted from AlphaFold predictions.

Relevant References:
Heo, L.; Feig, M. Multi-State Modeling of G-Protein Coupled Receptors at Experimental Accuracy. Proteins Struct. Funct. Bioinforma. 2022, 90, 1873–1885, doi:10.1002/PROT.26382.
He, X. heng; You, C. zhao; Jiang, H. liang; Jiang, Y.; Xu, H.E.; Cheng, X. AlphaFold2 versus Experimental Structures: Evaluation on G Protein-Coupled Receptors. Acta Pharmacol. Sin. 2022 441 2022, 44, 1–7, doi:10.1038/s41401-022-00938-y.
Díaz-Rovira, A.M.; Martín, H.; Beuming, T.; Díaz, L.; Guallar, V.; Ray, S.S. Are Deep Learning Structural Models Sufficiently Accurate for Virtual Screening? Application of Docking Algorithms to AlphaFold2 Predicted Structures. J. Chem. Inf. Model. 2023, 63, 1668–1674, doi:10.1021/ACS.JCIM.2C01270/ASSET/IMAGES/LARGE/CI2C01270_0003.JPEG.
Scardino, V.; Di Filippo, J.I.; Cavasotto, C.N. How Good Are AlphaFold Models for Docking-Based Virtual Screening? iScience 2022, 26, doi:10.1016/J.ISCI.2022.105920.
Cavasotto, C.N.; Di Filippo, J.I.; Scardino, V. Lessons Learnt from Machine Learning in Early Stages of Drug Discovery. Expert Opin. Drug Discov. 2024, 19, 631–633, doi:10.1080/17460441.2024.2354279

Please let us know where this is sufficient to accept our manuscript.

Academic Editor Evaluation Round 2

The authors provide compelling reasons for choosing template-based protein modeling over de novo methods like AlphaFold. Aside from this, the manuscript is ready for acceptance.

Second Academic Editor Evaluation

In my opinion, all concerns raised by the reviewers have now been addressed. The revisions have strengthened the paper, and it appears ready for publication.